# Low-Velocity Impact Behavior of Sandwich Plates with FG-CNTRC Face Sheets and Negative Poisson’s Ratio Auxetic Honeycombs Core

**DOI:** 10.3390/polym14142938

**Published:** 2022-07-20

**Authors:** Chunhao Yang, Wuning Ma, Zhendong Zhang, Jianlin Zhong

**Affiliations:** School of Mechanical Engineering, Nanjing University of Science and Technology, Nanjing 210094, China; kkmwn@163.com (W.M.); zzd1157@163.com (Z.Z.); zhongjianlin@njust.edu.cn (J.Z.)

**Keywords:** FG-CNTRC, auxetic honeycomb core, negative Poisson’s ratio, low-velocity impact

## Abstract

The combination of auxetic honeycomb and CNT reinforcement composite is expected to further improve the impact protection performance of sandwich structures. This paper studies the low-velocity impact response of sandwich plates with functionally graded carbon nanotubes reinforced composite (FG-CNTRC) face sheets and negative Poisson’s ratio (NPR) auxetic honeycomb core. The material properties of FG-CNTRC were obtained by the rule of mixture theory. The auxetic honeycomb core is made of Ti-6Al-4V. The governing equations are derived based on the first-order shear deformation theory and Hamilton’s principle. The nonlinear Hertz contact law is used to calculate the impact parameters. The Ritz method with Newmark’s time integration schemes is used to solve the response of the sandwich plates. The (20/−20/20)s, (45/−45/45)s and (70/−70/70)s stacking sequences of FG-CNTRC are considered. The effects of the gradient forms of FG-CNTRC surfaces, volume fractions of CNTs, impact velocities, temperatures, ratio of plate length, width and thickness of surface layers on the value of the plate center displacement, the recovery time of deformation, contact force and contact time of low-velocity impact were analyzed in detail.

## 1. Introduction

As the “Nanometer” material science, typified by carbon nanotubes (CNTs), develops, the widespread use of CNTs reinforcement composite (CNTRC) has brought changes to the sensor, intelligent medical and shelter structure fields [1,2,3]. The CNTs could improve the mechanical properties of composite and are remarkable as an ideal reinforcement. Shen [4] introduced functionally graded properties into CNTRC by designing the volume fraction of CNTs along the thickness direction, which avoids material properties suffering degradation due to the high levels of CNTs. Then, Kwon et al. [5] successfully made FG-CNTRC using powder metallurgy technology. At this point, large numbers of studies on the buckling [6,7,8,9,10,11,12,13,14,15,16,17] and vibration [18,19,20,21,22,23,24,25,26,27,28,29,30,31] analyses of FG-CNTRC structures have been carried out. Because of the low-velocity impact during the manufacture, installation use and maintenance, the inside structure of composite could be damaged and the lifting capacity will decrease and even fail. Therefore, studies on the low-velocity impact of FG-CNTRC were also carried out [32,33,34,35,36,37,38,39,40,41,42].

Most natural materials have the properties of expanding (contracting) laterally when compressed (stretched) longitudinally, which can be defined as positive Poisson’s ratio materials. In recent years, auxetic material has generated a lot of interest among researchers due to the negative Poisson’s ratio (NPR) properties [43,44,45]. Re-entrant [46], chiral [47] and other various materials have been proposed. Due to the outstanding performance on energy absorption [48,49,50], crashworthiness [51,52], and low-velocity impact resistance [53,54], auxetic material has been increasingly applied in biological medicine, photonics, energy storage, thermal management, and acoustic areas [55]. As an ideal core of sandwich structures, auxetic material could be used in shield structures in aerospace and civil engineering. Therefore, the nonlinear mechanical response of the sandwich structure with an auxetic honeycomb core [56,57] was analyzed by Li, Shen, and Wang [58,59,60,61,62,63,64]. Wan et al. [65] analyzed the uniaxial compression or expanded properties of auxetic honeycombs. Grima et al. [66] proposed a hexagonal honeycomb with zero Poisson’s ratios. Assidi and Ganghoffer [67] represented a composite with auxetic behavior and proved that the overall NPR could improve the mechanical properties. Grujicic et al. [68] focused on the sandwich structures with an auxetic hexagonal core and built the multi-physics model of fabrication and dynamic performance. Liu et al. [69] investigated the propagation of waves in a sandwich plate with a periodic composite core. Qiao and Chen [70] analyzed the impact response of auxetic double arrowhead honeycombs. Zhang et al. [71] analyzed the in-plane dynamic crushing behaviors and energy-absorbed characteristics of NPR honeycombs with cell microstructure. Zhang et al. [72] analyzed the dynamic mechanical and impact response on yarns with helical auxetic properties.

There are two main methods to propose auxetic structures: the first is using auxetic material as the core of sandwich plate [55]; and the second is changing the stacking sequence and orientation of laminate [73,74]. To realize a larger NPR value using the second method requires not only a specific stacking sequence but also a highly anisotropic properties of each ply [75]. Due to the mechanical properties of CNTs, the longitudinal elastic modulus $E_{11}$ of CNTRC is much larger than the transverse elastic modulus $E_{22}$ and large NPR properties can be proposed by designing the stacking sequence of CNTRC laminate. Then, Shen et al. [45,76] introduced the NPR property to the FG-CNTRC laminate and analyzed the nonlinear bending and free vibration response. Yang, Huang, and Shen [77,78], as well as Yu and Shen [79] analyzed the effects of an out-of-plane NPR property on large amplitude vibration and nonlinear bending of the FG-CNTRC laminated beam and plate. Fan, Wang [80] and Huang et al. [81,82] analyzed the dynamic response of the auxetic FG-CNTRC.

The combination of auxetic honeycomb and CNT reinforcement composite is expected to further improve the impact protection performance of sandwich structures. This paper studies the low-velocity impact response of the sandwich plates with functionally graded carbon nanotubes reinforced composite (FG-CNTRC) face sheets and a negative Poisson’s ratio (NPR) auxetic honeycomb core. The rule of mixture theory was used to calculate the material properties of FG-CNTRC with the PmPV matrix and CNTs reinforcement, while the effective Poisson’s ratio was obtained by laminate plate theory (Section 2.2). The NPR honeycomb core was made of Ti-6Al-4V (Section 2.3). The first-order shear deformation theory and Hamilton’s principle were used to describe the governing equations of the plate (Section 3.1). The nonlinear Hertz contact law was used to calculate the impact parameters (Section 3.2). The Ritz method with Newmark’s time integration schemes was used to solve the response of the sandwich plate (Section 3.3). After verifying the model, the (20/−20/20)s, (45/−45/45)s and (70/−70/70)s three kinds of stacking sequence of FG-CNTRC surfaces were considered. The effects of gradient forms of FG-CNTRC surfaces, volume fractions of CNTs, impact velocities, temperatures, ratio of plate length and the width and thickness of surface layers on low-velocity impact response were analyzed. The value of plate center displacement, recovery time of deformation, contact force and contact time were discussed in detail.

## 2. Modeling and Materials of Sandwich Plates

### 2.1. Modeling of Sandwich Plates

The sandwich plates with length *a*, width *b* and total thickness *h* are considered in this research, as shown in Figure 1. The face sheets with a thickness $h_{f}$ are FG-CNTRC-laminated structures composed of CNTRC layers with various volume fractions of CNTs. The auxetic core with a thickness of $h_{c}$ is the negative Poisson’s ratio honeycomb structure using isotropic titanium alloy (Ti-6Al-4V). A coordinate system (x,y,z) with (x,y) plane in the middle surface of the plate and *z* in the thickness direction is considered.

### 2.2. Materials of FG-CNTRC Face Sheets

The CNTRC layers with the poly(m-phenylenevinylene)-co-((2,5-dioctoxy-p-phenylene) vinylene) (PmPV) matrix are considered in this research. The material properties of the face sheets can be obtained based on the rule of mixture theory [4].
(1)$$\begin{array}{cc} \; & E_{11}=\eta_{1}V^{c}E_{11}^{c}+V^{m}E^{m},\;\rho =V^{c}\rho^{c}+V^{m}\rho^{m},\\ \; & \frac{\eta_{2}}{E_{22}}=\frac{V^{c}}{E_{22}^{c}}+\frac{V^{m}}{E^{m}},\;\frac{\eta_{3}}{G_{12}}=\frac{V^{c}}{G_{12}^{c}}+\frac{V^{m}}{G^{m}},\\ \; & \alpha_{11}=\frac{V^{c}E_{11}^{c}\alpha_{11}^{c}+V^{m}E^{m}\alpha^{m}}{V^{c}E_{11}^{c}+V^{m}E^{m}},\;\nu_{12}=V^{c}\nu_{12}^{c}+V^{m}\nu^{m}\\ \; & \alpha_{22}=\left(1+\nu_{12}^{c}\right)V^{c}\alpha_{22}^{c}+\left(1+\nu^{m}\right)V^{m}\alpha^{m}-\nu_{12}\alpha_{11} \end{array}$$
where the superscript *c* and *m* represent the material properties of CNTs and the matrix, respectively. *V* is the volume fraction, in which $V^{m}+V^{c}=1$. $\eta_{j}\left(j=1,2,3\right)$ is the efficiency parameters of CNTs. The values are shown in Table 1. *E*, *G*, ν, ρ and α are the elastic module, shear module, Poisson’s ratio, density and the thermal expansion of the materials, respectively. The (10, 10) SWCNTs are considered as the reinforcement in this research and the material properties are shown in Table 2. The material properties of the matrix PmPV are shown in Table 3.

The functionally graded properties of the CNTRC laminated structure are established according to the arrangement of CNTRC layers with the CNTs’ volume fractions of 0.11, 0.14 and 0.17. As shown in Figure 2, four types of FG-CNTRC, namely FG-V, FG-A, FG-O, FG-X and a uniformly distributed CNTRC with CNTs’ volume fractions of 0.14, namely UD, can be obtained. The laminated arrangement of FG-CNTRC can be expressed as
(2)$$\begin{array}{cc} \mathrm{FG}-V:\; & \left[{\left(0.17\right)}_{2}/{\left(0.14\right)}_{2}/{\left(0.11\right)}_{2}\right]\\ \mathrm{FG}-A:\; & \left[{\left(0.11\right)}_{2}/{\left(0.14\right)}_{2}/{\left(0.17\right)}_{2}\right]\\ \mathrm{FG}-O:\; & {\left[0.11/0.14/0.17\right]}_{s}\\ \mathrm{FG}-X:\; & {\left[0.17/0.14/0.11\right]}_{s} \end{array}$$

For an anisotropic laminated plate, the effective Poisson’s ratios $\nu_{13}^{e}$ and $\nu_{23}^{e}$ can be expressed as [44]
(3)$$\nu_{13}^{e}=-\frac{\left|\begin{array}{cc} A_{13} & B_{6-1}\\ B_{5-3} & D \end{array}\right|}{\left|\begin{array}{cc} A_{5-1} & B_{6-1}\\ B_{5-1} & D \end{array}\right|},\;\nu_{23}^{e}=\frac{\left|\begin{array}{cc} A_{23} & B_{6-2}\\ B_{5-3} & D \end{array}\right|}{\left|\begin{array}{cc} A_{5-2} & B_{6-2}\\ B_{5-2} & D \end{array}\right|}$$
where A, B and D are the stiffness matrix of the FG-CNTRC laminated surface. The aforementioned elements of the matrix are presented in Appendix A.

Combining the gradient forms of FG-CNTRC, the effective Poisson’s ratios could be calculated as shown in Figure 3. Three typical stacking sequences including (20/−20/20)s, (45/−45/45)s and (70/−70/70)s are considered to analyze the low-velocity impact response under various effective Poisson’s ratios.

### 2.3. Materials of Auxetic Honeycomb Core

The honeycomb core made of Ti-6Al-4V with negative Poisson’s ratio properties is considered in this research. The unit cell of the honeycomb is shown in Figure 4 and the material properties of the honeycomb core can be obtained by [56]
(4)$$\begin{array}{cc} \; & E_{1}^{h}=E_{Ti}{\left(\frac{t_{h}}{l_{h}}\right)}^{3}\frac{\cos\theta_{h}}{\left(h_{h}/l_{h}+\sin\theta_{h}\right)\sin^{2}\theta_{h}},\;E_{2}^{h}=E_{Ti}{\left(\frac{t_{h}}{l_{h}}\right)}^{3}\frac{\left(h_{h}/l_{h}+\sin\theta_{h}\right)}{\cos^{3}\theta_{h}}\\ \; & \nu_{12}^{h}=\frac{\cos^{2}\theta_{h}}{\left(h_{h}/l_{h}+\sin\theta_{h}\right)\sin\theta_{h}},\;G_{12}^{h}=E_{Ti}{\left(\frac{t_{h}}{l_{h}}\right)}^{3}\frac{\left(h_{h}/l_{h}+\sin\theta_{h}\right)}{{\left(h_{h}/l_{h}\right)}^{2}\left(1+2h_{h}/l_{h}\right)\cos\theta_{h}}\\ \; & G_{13}^{h}=G_{Ti}\frac{t_{h}}{l_{h}}\frac{\cos\theta_{h}}{h_{h}/l_{h}+\sin\theta_{h}},\;G_{23}^{h}=G_{Ti}\frac{t_{h}}{l_{h}}\frac{1+2\sin^{2}\theta_{h}}{2\cos\theta_{h}\left(h_{h}/l_{h}+\sin\theta_{h}\right)},\\ \; & \rho^{h}=\rho_{Ti}\frac{t_{h}/l_{h}\left(h_{h}/l_{h}+2\right)}{2\cos\theta_{h}\left(h_{h}/l_{h}+\sin\theta_{h}\right)} \end{array}$$
where the superscript *h* and subscript Ti represent the material properties of honeycomb and Ti-6Al-4V, respectively. $l_{h}$ represents the length of the inclined cell rib; $t_{h}$ represents the thickness of the cell rib; $h_{h}$ represents the length of the vertical cell rib; and $\theta_{h}$ represents the inclined angle. The original properties of the honeycomb can be controlled by the parameters above. The material properties of the Ti-6Al-4V are mentioned in Table 4.

## 3. Computational Methods

### 3.1. Governing Equations

The first-order shear deformation theory is used to describe the sandwich plate with length *a*, width *b* and thickness *h*, as shown in Figure 1. The displacement field $\left(\bar{u},\bar{v},\bar{w}\right)$ can be expressed as
(5)$$\begin{array}{cc} \; & \bar{u}\left(x,y,z,t\right)=u\left(x,y,t\right)+z\phi_{x}\left(x,y,t\right)\\ \; & \bar{v}\left(x,y,z,t\right)=v\left(x,y,t\right)+z\phi_{y}\left(x,y,t\right)\\ \; & \bar{w}\left(x,y,z,t\right)=w\left(x,y,t\right) \end{array}$$
where *u*, *v* and *w* are the translation displacement components at the mid-plane in the *x*, *y* and *z* directions, respectively. $\phi_{x}$ and $\phi_{y}$ denote the rotation of the normal to the mid-plane along the *y* axis and *x* axis, respectively. The relationship between strain and displacement can be expressed as
(6)$$\begin{array}{cc} \varepsilon & =\varepsilon^{0}+z\kappa^{0}\\ \gamma & =\gamma^{0} \end{array}$$
where
(7)$$\begin{array}{l} \varepsilon =\left[\begin{array}{c} \varepsilon_{xx}\\ \varepsilon_{yy}\\ \gamma_{xy} \end{array}\right],\;\varepsilon^{0}=\left[\begin{array}{cc} \; & \frac{\partial u}{\partial x}\\ \; & \frac{\partial v}{\partial y}\\ \frac{\partial v}{\partial x} & +\frac{\partial u}{\partial y} \end{array}\right],\;\kappa^{0}=\left[\begin{array}{cc} \; & \frac{\partial \phi_{x}}{\partial x}\\ \; & \frac{\partial \phi_{y}}{\partial y}\\ \frac{\partial \phi_{y}}{\partial x} & +\frac{\partial \phi_{x}}{\partial y} \end{array}\right],\\ \gamma =\left[\begin{array}{c} \gamma_{yz}\\ \gamma_{xz} \end{array}\right],\;\gamma^{0}=\left[\begin{array}{c} \phi_{y}+\frac{\partial w}{\partial y}\\ \phi_{x}+\frac{\partial w}{\partial x} \end{array}\right]. \end{array}$$

Considering the temperature effect, the stress component based on a linear constitutive relationship can be written as
(8)$$\left\{\begin{array}{c} \sigma_{xx}\\ \sigma_{yy}\\ \tau_{xy}\\ \tau_{yz}\\ \tau_{xz} \end{array}\right\}=\left[\begin{array}{ccccc} {\bar{Q}}_{11} & {\bar{Q}}_{12} & 0 & 0 & 0\\ {\bar{Q}}_{21} & {\bar{Q}}_{22} & 0 & 0 & 0\\ 0 & 0 & {\bar{Q}}_{66} & 0 & 0\\ 0 & 0 & 0 & {\bar{Q}}_{44} & 0\\ 0 & 0 & 0 & 0 & {\bar{Q}}_{55} \end{array}\right]\left(\left\{\begin{array}{c} \varepsilon_{xx}\\ \varepsilon_{yy}\\ \gamma_{xy}\\ \gamma_{yz}\\ \gamma_{xz} \end{array}\right\}-\left\{\begin{array}{c} \alpha_{11}\\ \alpha_{22}\\ 0\\ 0\\ 0 \end{array}\right\}\Delta T\right)$$
where ΔT is the temperature change and the transformed stiffness $\bar{Q}$ can be calculated by
(9)$$\left[\begin{array}{c} {\bar{Q}}_{11}\\ {\bar{Q}}_{12}\\ {\bar{Q}}_{22}\\ {\bar{Q}}_{16}\\ {\bar{Q}}_{26}\\ {\bar{Q}}_{66} \end{array}\right]=\left[\begin{array}{cccc} c^{4} & 2c^{2}s^{2} & s^{4} & 4c^{2}s^{2}\\ c^{2}s^{2} & c^{4}+s^{4} & c^{2}s^{2} & -4c^{2}s^{2}\\ s^{4} & 2c^{2}s^{2} & c^{4} & 4c^{2}s^{2}\\ c^{3}s & \left(cs^{3}-c^{3}s\right) & -cs^{3} & -2cs\left(c^{2}-s^{2}\right)\\ cs^{3} & \left(c^{3}s-cs^{3}\right) & -c^{3}s & 2cs\left(c^{2}-s^{2}\right)\\ c^{2}s^{2} & -2c^{2}s^{2} & c^{2}s^{2} & {\left(c^{2}-s^{2}\right)}^{2} \end{array}\right]\left[\begin{array}{c} Q_{11}\\ Q_{12}\\ Q_{22}\\ Q_{66} \end{array}\right]\;\left[\begin{array}{c} {\bar{Q}}_{44}\\ {\bar{Q}}_{45}\\ {\bar{Q}}_{55} \end{array}\right]=\left[\begin{array}{cc} c^{2} & s^{2}\\ -cs & cs\\ s^{2} & c^{2} \end{array}\right]\left[\begin{array}{c} Q_{44}\\ Q_{55} \end{array}\right]$$
where *s* and *c* are the sin and cos of the lamination angle against the *x* axis of the plate. Furthermore, the stiffness parameters can be given as
(10)$$\begin{array}{cc} \; & Q_{11}=\frac{E_{11}}{1-v_{12}v_{21}},\;Q_{22}=\frac{E_{22}}{1-v_{12}v_{21}},\;Q_{12}=\frac{v_{21}E_{11}}{1-v_{12}v_{21}}\\ \; & Q_{44}=G_{23},\;Q_{55}=G_{13},\;Q_{66}=G_{12} \end{array}$$

The strain energy of the sandwich plate $U_{p}$ can be expressed as
(11)$$U_{p}=\frac{1}{2}\int_{\Omega }{\bar{\varepsilon }}^{T}S\bar{\varepsilon }d\Omega$$
where $\bar{\varepsilon }={\left(\varepsilon^{0},\kappa^{0},\gamma^{0}\right)}^{T}$ is the strain matrix, S is the material constant matrix and
(12)$$S=\left[\begin{array}{ccc} A & B & 0\\ B & D & 0\\ 0 & 0 & A_{s} \end{array}\right]=\left[\begin{array}{cccccccc} A_{11} & A_{12} & A_{16} & B_{11} & B_{12} & B_{16} & 0 & 0\\ A_{12} & A_{22} & A_{26} & B_{12} & B_{22} & B_{26} & 0 & 0\\ A_{16} & A_{26} & A_{66} & B_{16} & B_{26} & B_{66} & 0 & 0\\ B_{11} & B_{12} & B_{16} & D_{11} & D_{12} & D_{16} & 0 & 0\\ B_{12} & B_{22} & B_{26} & D_{12} & D_{22} & D_{26} & 0 & 0\\ B_{16} & B_{26} & B_{66} & D_{16} & D_{26} & D_{66} & 0 & 0\\ 0 & 0 & 0 & 0 & 0 & 0 & A_{44}^{s} & A_{45}^{s}\\ 0 & 0 & 0 & 0 & 0 & 0 & A_{45}^{s} & A_{55}^{s} \end{array}\right]$$
where $A,B,D,A_{s}$ are the matrices of the plate stiffness, which can be calculated by
(13)$$\left(A,B,D\right)=\sum_{k=1}^{N}\int_{h_{k-1}}^{h_{k}}{\left(\bar{Q}\right)}_{k}\left(1,z,z^{2}\right)dz,\;A_{s}=K_{s}\sum_{k=1}^{N}\int_{h_{k-1}}^{h_{k}}{\left(\bar{Q}\right)}_{k}dz$$
where the transverse shear correction coefficient $K_{s}$ can be calculated by
(14)$$K_{s}=\left\{\begin{array}{c} \frac{5}{6},\;isotropic\;material\\ \frac{5}{6-\nu_{1}V_{1}-\nu_{2}V_{2}},\;functionally\;graded\;material \end{array}\right.$$
where ν and *V* are the Poisson’s ratios and volume fraction of each material in the entire cross-section. The kinetic energy of the sandwich plate T can be obtained by
(15)$$T=\frac{1}{2}\int_{\Omega }\int_{-h/2}^{h/2}\rho \left(z\right)\left({\overset{∸}{u}}^{2}+{\overset{∸}{v}}^{2}+{\overset{∸}{w}}^{2}\right)dz\;d\Omega$$

The external virtual work δW can be obtained by
(16)$$\delta W=F_{c}\left(t\right)\delta \mu$$
where $F_{c}\left(t\right)$ is the contact force between the plate and the impactor, and μ is the deflection of the sandwich plate. Then, the total energy function based on Hamilton’s principle can be expressed as
(17)$$\int_{0}^{t}\left(\delta U_{p}-\delta T-\delta W\right)dt=0$$

The boundary conditions for the clamped of the plate edge can be expressed as
(18)$$u=0,v=0,w=0,\phi_{x}=0,\phi_{y}=0$$

### 3.2. Low-Velocity Impact Response

Based on the nonlinear Hertz contact law, the contact force $F_{c}\left(t\right)$ between the sandwich plate and a steel ball can be obtained by [83]
(19)$$F_{c}\left(t\right)=\left\{\begin{array}{c} K_{c}\mu^{\frac{3}{2}}\left(t\right)\;\mathrm{loading}\\ F_{cm}{\left(\frac{\mu }{\mu_{m}}\right)}^{\frac{5}{2}}\;\mathrm{unloading} \end{array}\right.$$
where $\mu =w_{i}-w_{p}$ is the deflection of the sandwich plate, and $w_{i},w_{p}$ refers to the displacement of the impactor and plate center, respectively. The subscript *m* refers to the maximum value of the variables. $K_{c}$ is the contact coefficient, which can be expressed as [83],
(20)$$K_{c}=\frac{4}{3}{\left(\frac{1-\nu_{i}^{2}}{E_{i}}+\frac{1}{E_{2}}\right)}^{-1}\sqrt{r_{i}}$$
where $E_{i},\nu_{i},r_{i}$ are the elasticity modulus, Poisson’s ratios and the radius of the impactor, respectively. $E_{2}$ is the transverse elasticity modulus of the sandwich plate. The displacement of the impactor $w_{i}$ can be calculated by
(21)$$w_{i}=v_{i}t-\frac{1}{m_{i}}\int_{0}^{t}F_{c}\left(\tau \right)\left(t-\tau \right)d\tau$$
where $v_{i}$ and $m_{i}$ are the velocity and mass of the impactor, respectively. Then, the Equation (Equation 19) can be obtained by
(22)$${\left(\frac{F_{c}\left(t\right)}{K_{c}}\right)}^{2/3}=v_{i}t-\frac{1}{m_{i}}\int_{0}^{t}F_{c}\left(t-\tau \right)d\tau -w_{p}$$

### 3.3. Solution Procedure

The Ritz method is considered to deduce the governing equations of motion from the total energy function in the spatial domain, and the functions of the displacement field can be expressed as
(23)$$\begin{array}{cc} \; & u=\sum_{n=1}^{N}p_{n}^{u}\left(x,y\right)U_{n}\left(t\right)\\ \; & v=\sum_{n=1}^{N}p_{n}^{v}\left(x,y\right)V_{n}\left(t\right)\\ \; & w=\sum_{n=1}^{N}p_{n}^{w}\left(x,y\right)W_{n}\left(t\right)\\ \; & \phi_{x}=\sum_{n=1}^{N}p_{n}^{\phi_{x}}\left(x,y\right)\Phi_{xn}\left(t\right)\\ \; & \phi_{y}=\sum_{n=1}^{N}p_{n}^{\phi_{y}}\left(x,y\right)\Phi_{yn}\left(t\right) \end{array}$$
where $p_{n}\left(x,y\right)$ are the shape functions. n=1,2,⋯,N and *N* is the number of terms in the basis. $U_{n}\left(t\right)$, $V_{n}\left(t\right)$, $W_{n}\left(t\right)$, $\Phi_{xn}\left(t\right)$, $\Phi_{yn}\left(t\right)$ are the unknown coefficients chosen according to the boundary conditions. The shape functions of the polynomial are considered in this research [84,85].

The equations of motion of the sandwich plate and impactor can be obtained by
(24)$$\begin{array}{c} M\ddot{q}+Kq=F\\ m_{i}{\ddot{w}}_{i}+F_{c}=0 \end{array}$$
where q,M,K,F are the degrees of the freedom vector, mass matrix, stiffness matrix and impact load vector, respectively. Furthermore, the components of the mass matrix and the stiffness matrix are presented in Appendix B. The dot over the variable refers to the differentiation of that variable with respect to time. The Newmark’s time integration schemes is considered to solve the time-dependent equations after assembling the process and implementing boundary conditions. By using Taylor series expansions, the $q_{t+\Delta t}$, ${\dot{q}}_{t+\Delta t}$ and ${\ddot{q}}_{t+\Delta t}$ can be transformed into
(25)$$\begin{array}{cc} \; & q_{t+\Delta t}=q\left(t\right)+\Delta t{\dot{q}}_{t}+\frac{1}{2}\Delta t^{2}{\ddot{q}}_{t}-\frac{1}{2}\beta_{2}\Delta t^{2}{\ddot{q}}_{t}+\frac{1}{2}\beta_{2}\Delta t^{2}{\ddot{q}}_{t+\Delta t}\\ \; & {\dot{q}}_{t+\Delta t}={\dot{q}}_{t}+\Delta t{\ddot{q}}_{t}-\beta_{1}\Delta t{\ddot{q}}_{t}+\beta_{1}\Delta t{\ddot{q}}_{t+\Delta t}\\ \; & {\ddot{q}}_{t+\Delta t}=\frac{2}{\beta_{2}\Delta t^{2}}\left(q_{t+\Delta t}-q_{t}\right)-\frac{2}{\beta_{2}\Delta t^{2}}{\dot{q}}_{t}-\frac{1}{\beta_{2}}{\ddot{q}}_{t}+{\ddot{q}}_{t} \end{array}$$

Substituting Equation (Equation 25) into Equation (Equation 24): (26)$$\left(\frac{2}{\beta_{2}\Delta t^{2}}M+K\right)q_{t+\Delta t}=F_{t+\Delta t}+M\left(\frac{2}{\beta_{2}\Delta t^{2}}q_{t}+\frac{2}{\beta_{2}\Delta t}{\dot{q}}_{t},+\left(\frac{1}{\beta_{2}}-1\right){\ddot{q}}_{t}\right)$$
where the Newmark’s parameters $\beta_{1}=0.5$ and $\beta_{2}=0.5$ are considered in this research according to the Newmark β-method.

## 4. Results and Discussion

### 4.1. Validation Studies

To validate the calculation method, the relative examples of Refs. [38,86] are considered by contrast. The parameters of the plate are set to 1 m in length, 1 m in width and 0.01 m in thickness. The gradient form is UD while the $V^{c}$ is 0.28. The parameters of the impactor are set as a mass of 0.5 kg and a radius of 0.25 m. The working conditions are a temperature of 300 K and an initial impact velocity of 3 m/s. The displacement–time curve comparative result is shown in Figure 5. It can be inferred that the results are in good agreement. The maximum displacement and contact time error could be accepted for analysis.

In order to validate the equivalent layer model for the relative soft honeycomb core, a full-scale finite element simulation with an auxetic honeycomb core model was performed in contrast using the ABAQUS software, as shown in Figure 6. The sandwich structure with 0.5 mm thickness Ti-6Al-4V face sheets and auxetic honeycomb core was considered. The parameters of honeycomb core were set as: thickness $h_{c}$ = 23 mm; length of inclined cell rib $l_{h}$ = 5 mm; length of the vertical cell rib $h_{h}$ = 10 mm; and inclined angle $\theta_{h}$ = −40${}^{o}$. The second-order accuracy S4R elements were used to mesh the structure. Moreover, the meshes of face sheets are designed to share nodes with cores along the two interfaces, indicating the perfectly adhered to assumption. The impactor was set as an analytically rigid body ball with radius 10 mm. Furthermore, the mass was calculated according to the density 7.8 g/cm${}^{3}$. The general contact method with frictionless property was used to define the contact behavior. The initial impact velocity was 3 m/s, using predefined fields. All six degrees of freedoms of the boundary nodes were constrained to simulate clamped boundary conditions. The displacement–time curve comparative result is shown in Figure 7. It can be inferred that the results are in good agreement and the equivalent layer model could be used for the present research.

To be sure, the modeling method based on continuum mechanics theory in this paper was verified. The molecular dynamic theories or nano-scale continuum modeling is a more accurate simulation method for nanomaterials such as SCNT. However, this research focuses on the qualitative study of each parameter on the structural impact response, and the continuum mechanics theory can be used to show the trend of response after verification.

### 4.2. Parameter Studies

After verifying the model and computing method of this research, we focus on the (20/−20/20)s, (45/−45/45)s and (70/−70/70)s stacking sequences of the FG-CNTRC surface, the function gradient, volume fraction of CNTs, impact velocity, temperature, length/width ratio and FG-CNTRC surface thickness effects on the low-velocity impact response of the sandwich plate with FG-CNTRC face sheets and NPR auxetic honeycomb core are analyzed. The plate center displacement $w_{p}$, recovery time of deformation $t_{r}$, contact force $F_{c}$ and contact time $t_{c}$ are considered in detail. The initial parameters of the sandwich plate structure and boundary conditions are set as:Sandwich plate—length/width ratio a/b = 1, total thickness *h* = 25.4 mm;FG-CNTRC surface—thickness $h_{s}$ = 1.2 mm, gradient form FG-V;Honeycomb core—thickness $h_{c}$ = 23 mm, length of inclined cell rib $l_{h}$ = 5 mm, length of the vertical cell rib $h_{h}$ = 10 mm, inclined angle $\theta_{h}$ = −40${}^{\circ }$;Calculate conditions—temperature *T* = 300 K, impact velocity $v_{i}$ = 2 m/s, boundary conditions clamped.

#### 4.2.1. Gradient Forms of FG-CNTRC Surfaces

The low-velocity impact of gradient forms FG-V, FG-A, FG-X, FG-O and UD are considered. The plate center displacement of the three stacking sequences are shown in Figure 8. The (20/−20/20)s ply has the largest plate center displacement $w_{p}$, reaches the maximum value first and has the shortest recovery time of deformation $t_{r}$. The (45/−45/45)s ply has the smallest plate center displacement $w_{p}$. The (70/−70/70)s ply has the longest recovery time of deformation $t_{r}$. The value of the plate center displacement $w_{p}$, recovery time of deformation $t_{r}$, contact force $F_{c}$ and contact time $t_{c}$ are shown in Table 5 in detail. The UD form of (20/−20/20)s ply and (70/−70/70)s ply has the largest $w_{p}$, smallest $F_{c}$ and longest $t_{r}$. The FG-O form of (20/−20/20)s ply has the smallest $w_{p}$, largest $F_{c}$ and shortest $t_{r}$. While the FG-X form of (70/−70/70)s ply has the smallest $w_{p}$, largest $F_{c}$ and shortest $t_{r}$. The response of the (45/−45/45)s ply is more complicated. The UD form has the largest $w_{p}$ and longest $t_{r}$. The FG-X form has the largest $F_{c}$ and shortest $t_{r}$. The FG-O form has the smallest $w_{p}$. The FG-V form has the smallest $F_{c}$. The contact time $t_{c}$ of each gradient forms are nearly the same.

It is observed that the (45/−45/45)s ply with nearly zero Poisson’s ratio has the smallest $w_{p}$, and the (70/−70/70)s ply with the native $v_{23}^{e}$ has the smallest $F_{c}$. Within three stacking sequences and five gradient forms, (45/−45/45)s ply with FG-O type has the smallest $w_{p}$, while (70/−70/70)s ply with UD type has the smallest $F_{c}$. The percentage decrease is approximately 5% by changing the stacking sequence and gradient form of the surface sheets.

#### 4.2.2. Volume Fractions of CNTs

The 0.11, 0.14 and 0.17 volume fractions of CNTs are considered. The surface layer of this part of the research is set as uniform distribution. The plate center displacement are shown in Figure 9. The (20/−20/20)s ply has the largest plate center displacement $w_{p}$ and shortest recovery time of deformation $t_{r}$. The (45/−45/45)s ply has the smallest plate center displacement $w_{p}$ and the (70/−70/70)s ply has the longest recovery time of deformation $t_{r}$. According to Table 6, the response of three stacking sequences is similar. With the volume fractions of CNTs increasing, the plate center displacement $w_{p}$, recovery time of deformation $t_{r}$ and contact time $t_{c}$ decreases, while the contact force $F_{c}$ increases. It can be inferred that the contact stiffness increases with the volume fractions of CNTs increasing.

It is observed that increasing the stiffness of the sandwich structure by increasing the volume fraction of CNTs can lead to a reduction in the $w_{p}$ and an increase of the $F_{c}$. Furthermore, this phenomenon is more sensitive to (20/−20/20)s ply with a reduction in $w_{p}$ by approximately 6.4%.

#### 4.2.3. Impact Velocity

The impact velocity plays an important role in the impact response. Considering 1 m/s, 2 m/s and 3 m/s impact velocity, the plate center displacements of three stacking sequences are shown in Figure 10. The (20/−20/20)s ply has the largest plate center displacement $w_{p}$ and has the shortest recovery time of deformation $t_{r}$. The (45/−45/45)s ply has the smallest plate center displacement $w_{p}$. The (70/−70/70)s ply has the longest recovery time of deformation $t_{r}$. According to Table 7, with the increased impact velocity, the plate center displacement $w_{p}$ and the contact force $F_{c}$ increased, while the recovery time of deformation $t_{r}$ and contact time $t_{c}$ decreased.

It is observed that the three stacking sequences have a slight impact on the variable ratio of $w_{p}$ and $F_{c}$. Increasing the impact velocity from 1 m/s to 3 m/s can lead to an increase in the $w_{p}$ and $F_{c}$ by approximately 62.5% and 68%, respectively.

#### 4.2.4. Temperature

The low-velocity impact response of FG-CNTRC plates under various temperatures is the hotspot of its application under extreme conditions. The temperatures of 300 K, 400 K and 500 K are considered, as shown in Figure 11. Similarly to the result of various impact velocities, the (20/−20/20)s ply has the largest plate center displacement $w_{p}$ and has the shortest recovery time of deformation $t_{r}$. The (45/−45/45)s ply has the smallest plate center displacement $w_{p}$. The (70/−70/70)s ply has the longest recovery time of deformation $t_{r}$. According to Table 8, with the increased temperature, the plate center displacement $w_{p}$, recovery time of deformation $t_{r}$ and contact time $t_{c}$ increased, while the contact force $F_{c}$ decreased.

It is observed that the stiffness of the sandwich structure will reduce by increasing the temperature. From 300 K to 500 K, the $w_{p}$ will increase by approximately 8.4%.

#### 4.2.5. Ratio of Plate Length and Width

The length/width ratio a/b = 0.5, 1.0 and 2.0 are considered, as shown in Figure 12. The coupling between stacking sequence and a/b makes the low-velocity impact response complicated. The a/b = 2.0 has the largest plate center displacement $w_{p}$, while a/b = 0.5 is the smallest of all three stacking sequences. The responses are shown in Table 9 in detail. When a/b = 0.5, the (70/−70/70)s ply has the largest $w_{p}$ and smallest $F_{c}$, the (45/−45/45)s ply has the smallest $w_{p}$ and largest $F_{c}$. When a/b = 2.0, whilst the (45/−45/45)s ply has the largest $w_{p}$ and smallest $F_{c}$, the (20/−20/20)s ply has the smallest $w_{p}$ and largest $F_{c}$. However, the $t_{r}$ decreases at first and then increases with the increase in a/b. The $t_{c}$ increases with the increase in a/b. The results inferred that the ratio of plate length and width has a large influence on the low-velocity impact, which causes the nonlinear change phenomenon.

It is observed that the geometry scale has more influence on the impact response, due to the anisotropic honeycomb core. Using the honeycomb section as the long side of the structure can reduce the $F_{c}$.

#### 4.2.6. Thickness of Surface Layer

The thickness of the FG-CNTRC surface layer $h_{s}$ = 0.6 mm, 1.2 mm and 2.4 mm are considered, and the low-velocity impact response is shown in Figure 13. When $h_{s}$ = 1.2 mm and 2.4 mm, the stacking sequence has a large influence on the plate displacement $w_{p}$. According to Table 10, when $h_{s}$ = 0.6 mm, the (20/−20/20)s ply has the smallest $w_{p}$, largest $F_{c}$ and shortest $t_{r}$ and $t_{c}$. The (45/−45/45)s ply has the largest $w_{p}$, longest $t_{r}$ and $t_{c}$. The (70/−70/70)s ply has the smallest $F_{c}$. When $h_{s}$ = 2.4 mm, the (20/−20/20)s ply has the smallest $w_{p}$, largest $F_{c}$ and shortest $t_{r}$ and $t_{c}$. The (45/−45/45)s ply has the largest $w_{p}$, smallest $F_{c}$ and longest $t_{r}$ and $t_{c}$.

**Table 9 polymers-14-02938-t009:** Low-velocity impact response of a sandwich structure with various a/b.

Type	a/b	$w_{p}$ (mm)	$F_{c}$ (N)	$t_{r}$ (ms)	$t_{c}$ (ms)
	0.5	2.342	1147.677	9.380	3.900
(20/−20/20)s	1.0	2.522	1154.605	4.972	5.250
	2.0	2.624	1044.604	5.498	5.600
	0.5	2.275	1165.578	8.125	4.350
(45/−45/45)s	1.0	2.439	1162.674	5.306	5.650
	2.0	2.813	955.473	6.454	6.550
	0.5	2.354	1111.640	8.070	4.750
(70/−70/70)s	1.0	2.494	1105.591	5.879	5.750
	2.0	2.791	965.506	6.417	6.500

It is observed that increasing $h_{s}$ can lead to a reduction in the $w_{p}$ and an increase in the $F_{c}$ by increasing the stiffness of the structure.

## 5. Conclusions

In this research, a numerical method on the low-velocity impact response of the sandwich plate with an FG-CNTRC surface and NPR honeycomb core was proposed and verified. Three kinds of stacking sequences of FG-CNTRC, namely (20/−20/20)s, (45/−45/45)s and (70/−70/70)s, were considered. The effects of gradient forms of FG-CNTRC surfaces, volume fractions of CNTs, impact velocities, temperatures, the ratio of the plate length and the width and thickness of surface layers on the low-velocity impact response were analyzed. The results of the plate center displacement $w_{p}$, recovery time of deformation $t_{r}$, contact force $F_{c}$ and contact time $t_{c}$ show that:Gradient forms of FG-CNTRC surfaces:(20/−20/20)s ply—the UD form has the largest $w_{p}$, smallest $F_{c}$ and longest $t_{r}$; and the FG-O form has the smallest $w_{p}$, largest $F_{c}$ and shortest $t_{r}$;(45/−45/45)s ply—the UD form has the largest $w_{p}$ and longest $t_{r}$; the FG-X form has the largest $F_{c}$ and shortest $t_{r}$; the FG-O form has the smallest $w_{p}$; and the FG-V form has the smallest $F_{c}$;(70/−70/70)s ply—the UD form has the largest $w_{p}$, smallest $F_{c}$ and longest $t_{r}$; the FG-X form has the smallest $w_{p}$, largest $F_{c}$ and shortest $t_{r}$.Within three stacking sequences and five gradient forms, the (45/−45/45)s ply with FG-O type has the smallest $w_{p}$, while the (70/−70/70)s ply with the UD type has the smallest $F_{c}$. The percentage decrease is approximately 5% by changing the stacking sequence and gradient form of the surface sheets.Volume fractions of CNTs:The (20/−20/20)s ply has the largest $w_{p}$ and shortest $t_{r}$. The (45/−45/45)s ply has the smallest $w_{p}$ and the (70/−70/70)s ply has the longest $t_{r}$;The plate center displacement $w_{p}$, recovery time of deformation $t_{r}$ and contact time $t_{c}$ decreased, while the contact force $F_{c}$ increased with the increased volume fractions of CNTs.Increasing the volume fraction of CNTs from 0.11 to 0.17 can lead to a reduction in the $w_{p}$ and an increase in the $F_{c}$. Furthermore, this phenomenon is more sensitive to (20/−20/20)s ply with a reduction in $w_{p}$ by approximately 6.4%.Impact velocities:The (20/−20/20)s ply has the largest $w_{p}$ and has the shortest $t_{r}$. The (45/−45/45)s ply has the smallest $w_{p}$. The (70/−70/70)s ply has the longest $t_{r}$.The plate center displacement $w_{p}$ and contact force $F_{c}$ increased, while the recovery time of deformation $t_{r}$ and contact time $t_{c}$ decreased as the impact velocity increased.The three stacking sequences have a slight impact on the variable ratio of $w_{p}$ and $F_{c}$. Increasing the impact velocity from 1 m/s to 3 m/s can lead to an increase in the $w_{p}$ and $F_{c}$ of approximately 62.5% and 68%, respectively.Temperatures:The (20/−20/20)s ply has the largest $w_{p}$ and the shortest $t_{r}$. The (45/−45/45)s ply has the smallest $w_{p}$. The (70/−70/70)s ply has the longest $t_{r}$.The plate center displacement $w_{p}$, recovery time of deformation $t_{r}$ and contact time $t_{c}$ increased, while the contact force $F_{c}$ decreased as the temperature increased.The stiffness of the structure will reduce by increasing the temperature. From 300 K to 500 K, the $w_{p}$ will increase by approximately 8.4%.Ratio of plate length and width:(20/−20/20)s ply: a/b = 2.0 has the smallest $w_{p}$ and largest $F_{c}$.(45/−45/45)s ply: a/b = 0.5 has the smallest $w_{p}$ and largest $F_{c}$; a/b = 2.0 has the largest $w_{p}$ and smallest $F_{c}$.(70/−70/70)s ply: a/b = 0.5 has the largest $w_{p}$ and smallest $F_{c}$.The $t_{r}$ decreased at first and then increased as a/b increased.The $t_{c}$ increased as a/b increased.Due to the anisotropic honeycomb core, the geometry scale has more influence on the impact response. Using the honeycomb section as the long side of the structure can reduce the $F_{c}$.Thickness of surface layers:(20/−20/20)s ply: $h_{s}$ = 0.6 mm has the smallest $w_{p}$, largest $F_{c}$ and shortest $t_{r}$ and $t_{c}$; $h_{s}$ = 2.4 mm has the smallest $w_{p}$, largest $F_{c}$ and shortest $t_{r}$ and $t_{c}$.(45/−45/45)s ply: $h_{s}$ = 0.6 mm has the largest $w_{p}$, longest $t_{r}$ and $t_{c}$; $h_{s}$ = 2.4 mm has the largest $w_{p}$, smallest $F_{c}$ and longest $t_{r}$ and $t_{c}$.(70/−70/70)s ply: $h_{s}$ = 0.6 mm has the smallest $F_{c}$.Increasing $h_{s}$ can lead to a reduction in the $w_{p}$ and an increase in the $F_{c}$ by increasing the stiffness of the structure.

## Figures and Tables

**Figure 1 polymers-14-02938-f001:**
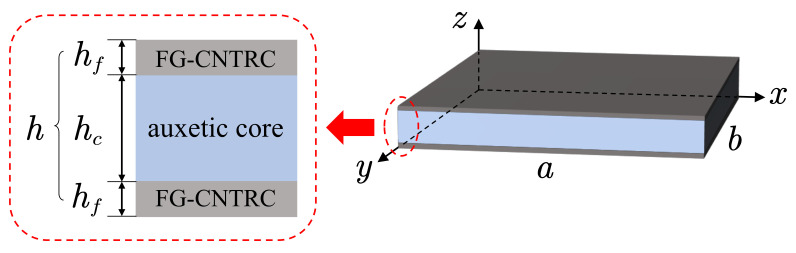
The sandwich plates with FG-CNTRC face sheets and auxetic honeycomb core.

**Figure 2 polymers-14-02938-f002:**
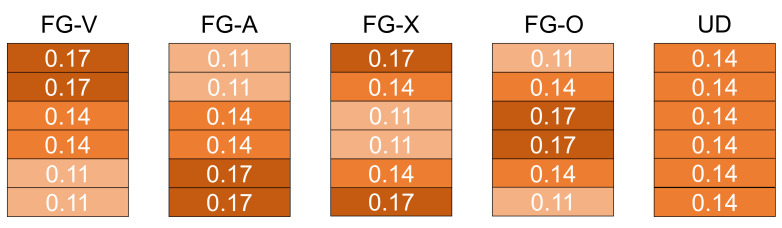
The CNTs’ volume fractions arrangement of five types of CNTRC laminate.

**Figure 3 polymers-14-02938-f003:**
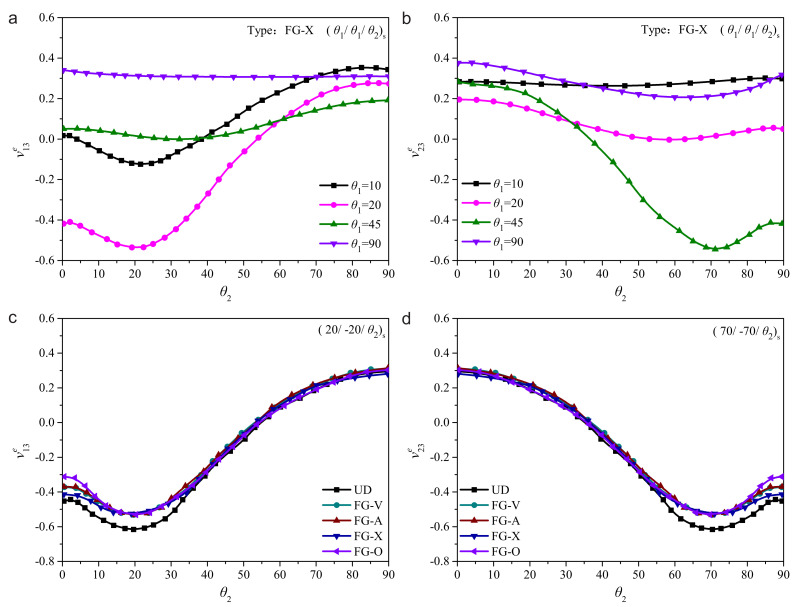
The effective Poisson’s ratios of FG-CNTRC laminated plates: (**a**) $\nu_{13}^{e}$ for ${\left(\theta_{1}/\theta_{1}/\theta_{2}\right)}_{s}$ laminates of type FG-X; (**b**) $\nu_{23}^{e}$ for ${\left(\theta_{1}/\theta_{1}/\theta_{2}\right)}_{s}$ laminates of type FG-X; (**c**) $\nu_{13}^{e}$ for (20/−20/$\theta_{2}{)}_{s}$ laminates; and (**d**) $\nu_{23}^{e}$ for (20/−20/$\theta_{2}{)}_{s}$ laminates.

**Figure 4 polymers-14-02938-f004:**
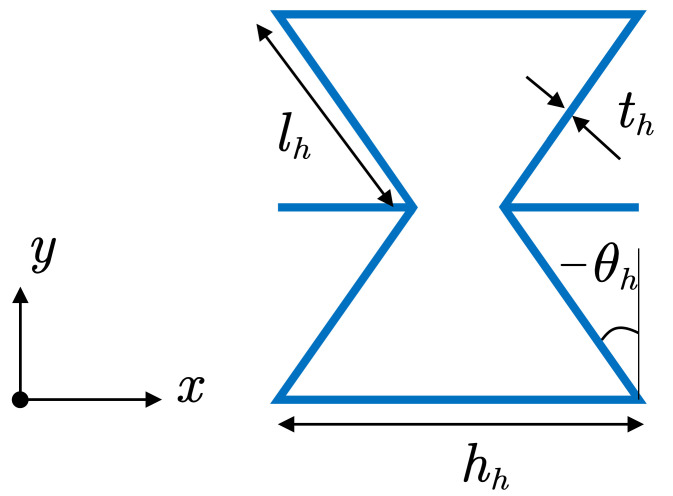
The structure of the auxetic honeycomb core.

**Figure 5 polymers-14-02938-f005:**
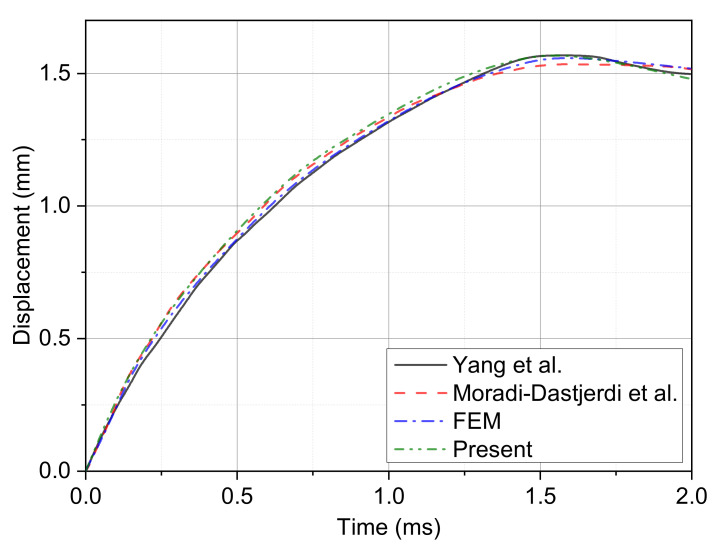
Comparison of the plate center displacement with the results obtained from the Refs. [38,86] and FEM method.

**Figure 6 polymers-14-02938-f006:**
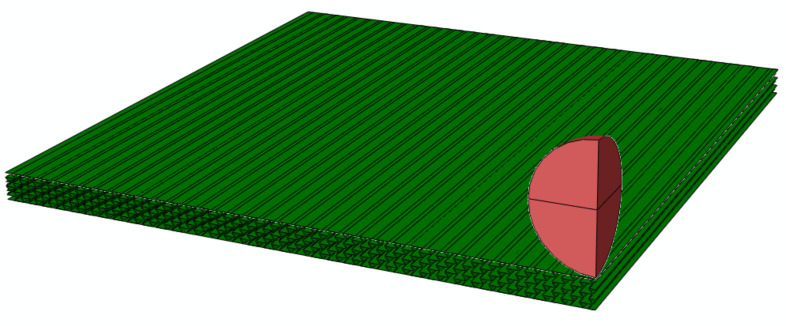
Low-velocity impact simulation in ABAQUS software.

**Figure 7 polymers-14-02938-f007:**
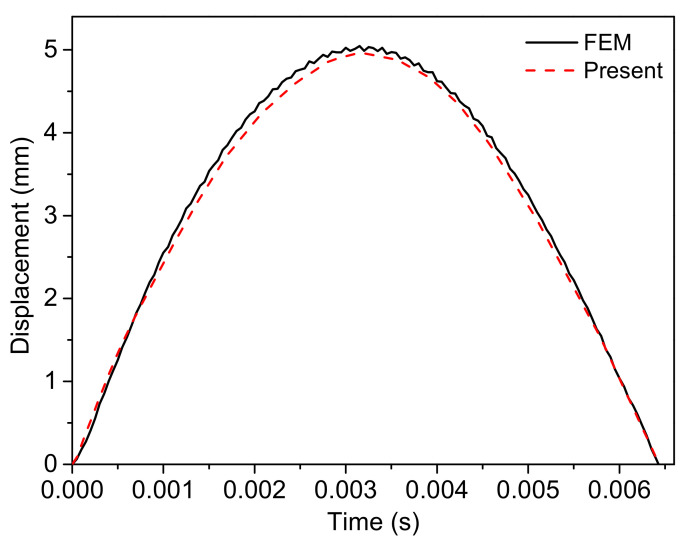
Comparison of the plate center displacement with the results obtained from FEM and present method.

**Figure 8 polymers-14-02938-f008:**
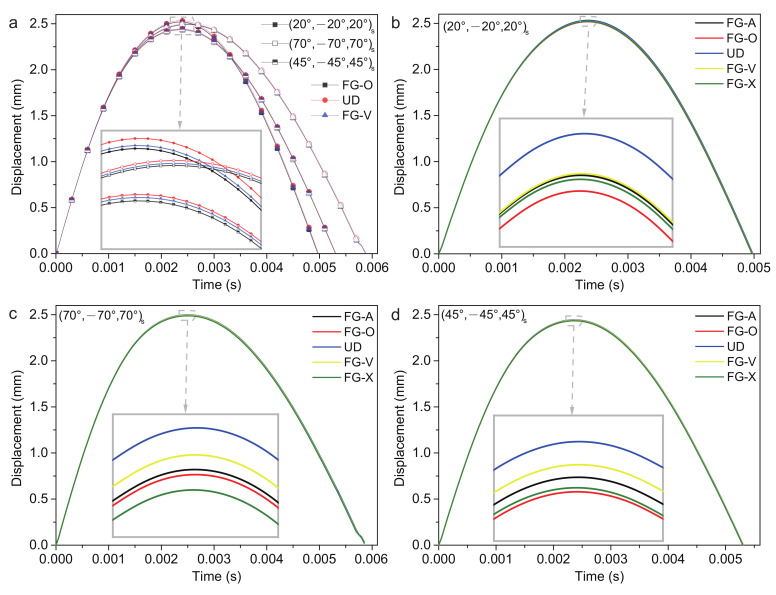
Plate center displacement response of a sandwich structure with various gradient forms: (**a**) FG-O, UD and FG-V face sheets plate; (**b**) (20/−20/20)s plate; (**c**) (70/−70/70)s plate; and (**d**) (45/−45/45)s plate.

**Figure 9 polymers-14-02938-f009:**
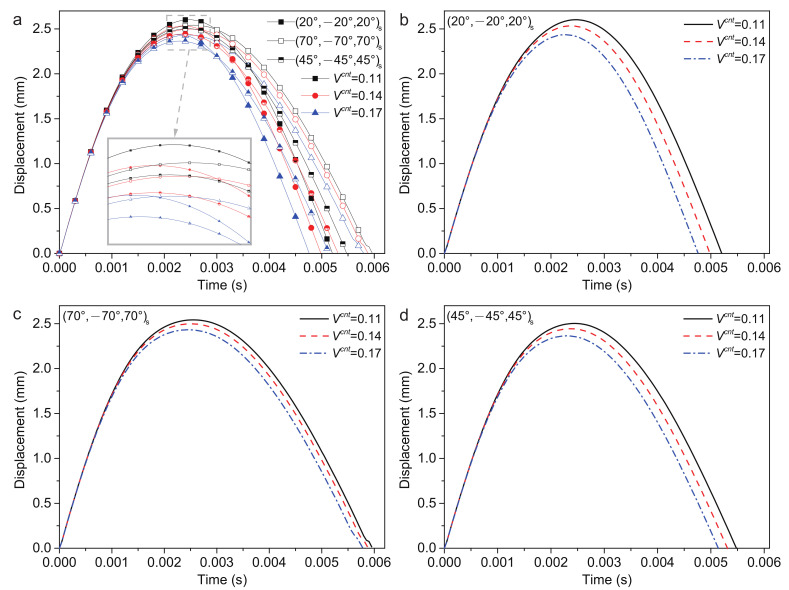
Plate center displacement response of the sandwich structure with various volume fractions of CNTs: (**a**) FG-O, UD and FG-V face sheets plate; (**b**) (20/−20/20)s plate; (**c**) (70/−70/70)s plate; (**d**) (45/−45/45)s plate.

**Figure 10 polymers-14-02938-f010:**
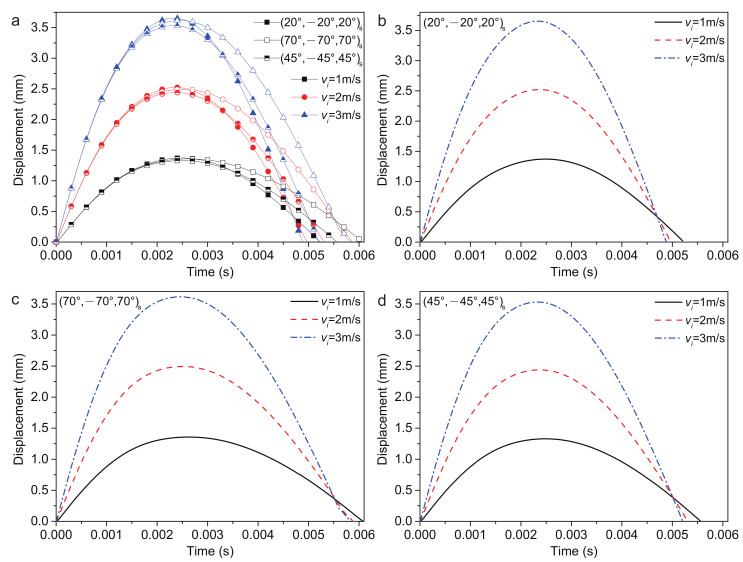
Plate center displacement response of the sandwich structure with various impact velocities: (**a**) FG-O, UD and FG-V face sheets plate; (**b**) (20/−20/20)s plate; (**c**) (70/−70/70)s plate; and (**d**) (45/−45/45)s plate.

**Figure 11 polymers-14-02938-f011:**
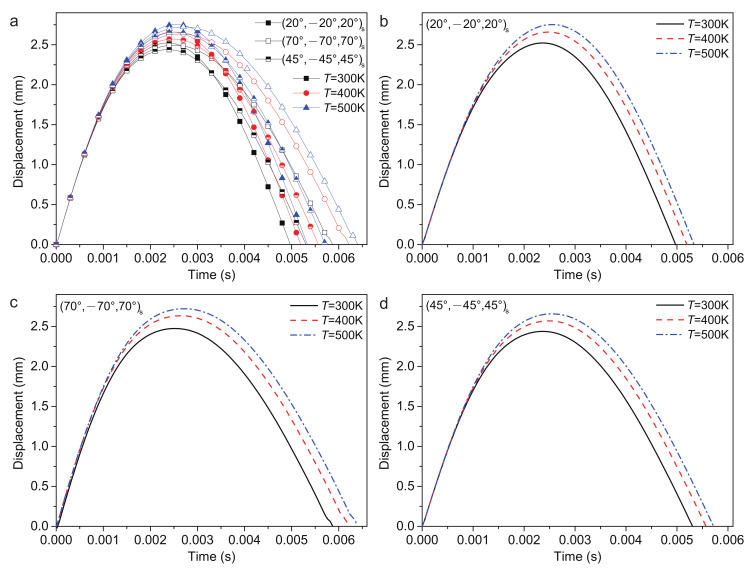
Plate center displacement response of the sandwich structure with various temperatures: (**a**) FG-O, UD and FG-V face sheets plate; (**b**) (20/−20/20)s plate; (**c**) (70/−70/70)s plate; and (**d**) (45/−45/45)s plate.

**Figure 12 polymers-14-02938-f012:**
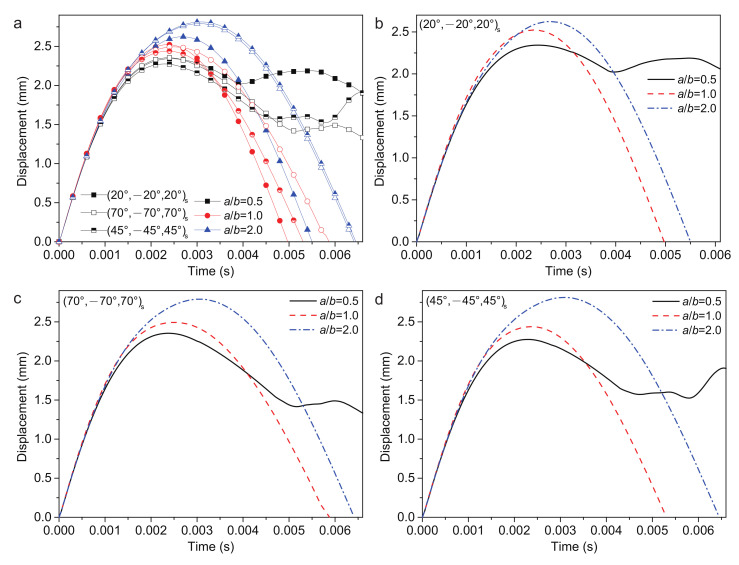
Plate center displacement response of the sandwich structure with various a/b: (**a**) FG-O, UD, and FG-V face sheets plate; (**b**) (20/−20/20)s plate; (**c**) (70/−70/70)s plate; and (**d**) (45/−45/45)s plate.

**Figure 13 polymers-14-02938-f013:**
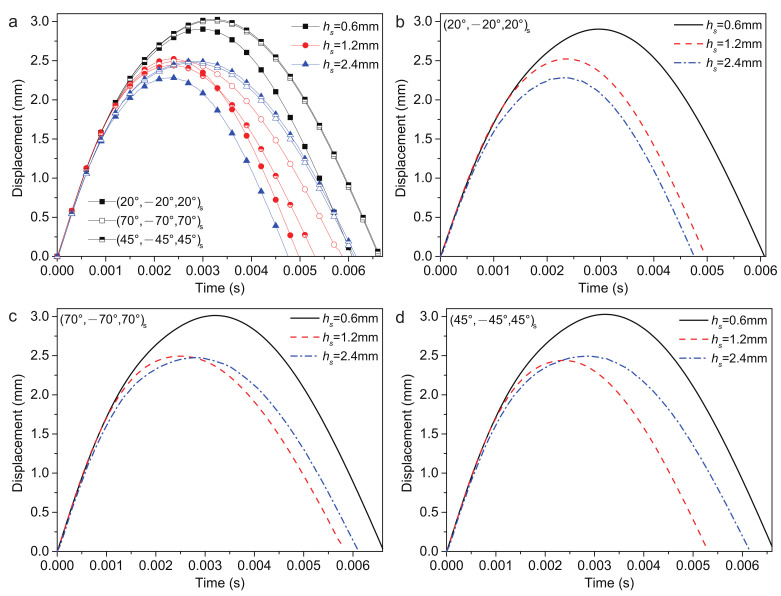
Plate center displacement response of the sandwich structure with various $h_{s}$: (**a**) FG-O, UD and FG-V face sheets plate; (**b**) (20/−20/20)s plate; (**c**) (70/−70/70)s plate; and (**d**) (45/−45/45)s plate.

**Table 1 polymers-14-02938-t001:** The efficiency parameter of CNTs [4].

$V^{\mathrm{cnt}}$	$\eta_{1}$	$\eta_{2}$	$\eta_{3}$
0.11	0.149	0.934	0.934
0.14	0.150	0.941	0.941
0.17	0.149	1.381	1.381

**Table 2 polymers-14-02938-t002:** The material properties of (10, 10) SWCNTs (tube radius = 0.68 nm, thickness = 0.067 nm, length = 9.26 nm, $\nu_{12}^{cnt}$ = 0.175) [10].

Temp (K)	$E_{11}^{\mathrm{cnt}}$ (TPa)	$E_{22}^{\mathrm{cnt}}$ (TPa)	$G_{12}^{\mathrm{cnt}}$ (TPa)	$\nu_{12}^{\mathrm{cnt}}$	$\alpha_{11}^{\mathrm{cnt}}$ (×10${}^{-6}$/K)	$\alpha_{22}^{\mathrm{cnt}}$ (×10${}^{-6}$/K)
300	5.6466	7.0800	1.9445	0.175	3.4584	5.1682
400	5.5308	6.9348	1.9643	0.175	4.1496	5.0905
500	5.4744	6.8641	1.9644	0.175	4.5361	5.0189

**Table 3 polymers-14-02938-t003:** The material properties of PmPV [76].

Temp (K)	$E^{\mathrm{pm}}$ (GPa)	$\nu^{\mathrm{pm}}$	$\alpha^{\mathrm{pm}}$ ($\times 10^{-6}$/K)
300	2.10	0.34	45.00
400	1.63	0.34	47.25
500	1.16	0.34	49.50

**Table 4 polymers-14-02938-t004:** The material properties of Ti-6Al-4V.

Material Properties	$E_{\mathrm{Ti}}$ (GPa)	$\nu_{12}$	$\rho_{\mathrm{Ti}}$ (g/cm${}^{3}$)
Ti-6Al-4V	122.56× (1–4.586 $\times \;10^{-4}T)$	0.342	4.43

**Table 5 polymers-14-02938-t005:** Low-velocity impact response of the sandwich structure with various gradient forms.

Type	Gradient Forms	$w_{p}$ (mm)	$F_{c}$ (N)	$t_{r}$ (ms)	$t_{c}$ (ms)
(20/−20/20)s	FG-A	2.522	1155.943	4.970	5.250
	FG-O	2.518	1156.313	4.966	5.250
	UD	2.534	1149.806	4.982	5.250
	FG-V	2.522	1154.605	4.972	5.250
	FG-X	2.521	1155.906	4.969	5.250
(45/−45/45)s	FG-A	2.436	1164.017	5.304	5.650
	FG-O	2.433	1164.171	5.303	5.650
	UD	2.444	1163.188	5.311	5.650
	FG-V	2.439	1162.674	5.306	5.650
	FG-X	2.434	1164.324	5.300	5.650
(70/−70/70)s	FG-A	2.491	1106.108	5.876	5.750
	FG-O	2.490	1106.045	5.875	5.750
	UD	2.498	1104.619	5.888	5.750
	FG-V	2.494	1105.591	5.879	5.750
	FG-X	2.488	1107.330	5.874	5.750

**Table 6 polymers-14-02938-t006:** Low-velocity impact response of a sandwich structure with various volume fraction of CNTs.

Type	Volume Fraction	$w_{p}$ (mm)	$F_{c}$ (N)	$t_{r}$ (ms)	$t_{c}$ (ms)
	0.11	2.602	1120.218	5.202	5.500
(20/−20/20)s	0.14	2.534	1149.936	4.978	5.250
	0.17	2.436	1196.062	4.762	5.000
	0.11	2.503	1134.566	5.478	5.800
(45/−45/45)s	0.14	2.444	1163.188	5.311	5.650
	0.17	2.365	1194.500	5.139	5.450
	0.11	2.436	1092.485	5.969	5.850
(70/−70/70)s	0.14	2.498	1104.619	5.880	5.750
	0.17	2.433	1121.323	5.783	5.600

**Table 7 polymers-14-02938-t007:** Low-velocity impact response of the sandwich structure with various impact velocities.

Type	Impact Velocity (m/s)	$w_{p}$ (mm)	$F_{c}$ (N)	$t_{r}$ (ms)	$t_{c}$ (ms)
	1	1.372	563.495	5.223	5.500
(20/−20/20)s	2	2.522	1154.605	4.972	5.250
	3	3.654	1784.333	4.881	5.150
	1	1.329	575.804	5.558	5.850
(45/−45/45)s	2	2.439	1162.674	5.306	5.650
	3	3.532	1777.370	5.201	5.350
	1	1.357	551.596	6.074	7.300
(70/−70/70)s	2	2.494	1105.591	5.879	5.750
	3	3.616	1706.855	5.848	7.200

**Table 8 polymers-14-02938-t008:** Low-velocity impact response of the sandwich structure with various temperatures.

Type	Temperature (K)	$w_{p}$ (mm)	$F_{c}$ (N)	$t_{r}$ (ms)	$t_{c}$ (ms)
	300	2.522	1154.605	4.972	5.250
(20/−20/20)s	400	2.659	1092.760	5.193	5.550
	500	2.753	1119.235	5.332	5.560
	300	2.439	1162.674	5.306	5.650
(45/−45/45)s	400	2.571	1104.613	5.570	5.950
	500	2.659	1098.925	5.714	6.100
	300	2.494	1105.591	5.879	5.750
(70/−70/70)s	400	2.635	1044.606	6.221	6.050
	500	2.723	1011.190	6.405	6.250

**Table 10 polymers-14-02938-t010:** Low-velocity impact response of the sandwich structure with various $h_{s}$.

Type	*h* (mm)	$w_{p}$ (mm)	$F_{c}$ (N)	$t_{r}$ (ms)	$t_{c}$ (ms)
	0.6	2.903	946.210	6.072	6.250
(20/−20/20)s	1.2	2.522	1154.605	4.970	5.250
	2.4	2.287	1746.733	4.746	4.850
	0.6	3.027	904.395	6.650	6.900
(45/−45/45)s	1.2	2.439	1162.674	5.302	5.650
	2.4	2.494	1209.172	6.149	6.350
	0.6	3.013	902.615	6.625	6.850
(70/−70/70)s	1.2	2.494	1105.591	5.877	5.750
	2.4	2.476	1469.982	6.102	6.300

## Data Availability

Not applicable.

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
