# Peer review of "Low-Velocity Impact Behavior of Sandwich Plates with FG-CNTRC Face Sheets and Negative Poisson’s Ratio Auxetic Honeycombs Core"

_polymers, 2022, doi:10.3390/polym14142938_

Round 1

Reviewer 1 Report

The present article is a numerical study of impact behaviour of sandwich plates with FG-CNTRC face sheets and negative Poisson’s ratio auxetic honeycombs core.

The abstract should be modified. In the abstract and the body text, Some phrases are exaggerated. The authors claim that they develop a numerical model or method to evaluate the impact-resistant of a honeycomb structure, however they used the classical laminated theory. I couldn’t find novelty in model development however, the idea to use CNT combination with honeycomb is acceptable.

The abbreviation word ‘FG-CNTRC’ in abstract should be defined first or use the full term.

They have used FEM software to evaluate the low-velocity response on the structure. This is a common mistake by many authors and to see the behaviour of nanoscale materials FE method can not be reliable. This is because of continuum mechanics theory behind this method. It is not correct to use continuum mechanics that is defined for calculation of micro to meso scale. Many publications have repeated this mistake and due to a lack of experiments, this way is continuing. I strongly recommend you use molecular dynamic theories or nano-scale continuum modelling (NCM)  when you are studying CNT structure. However, they can use FE models to see only the trend of parameter variation and they need to mention this in their manuscript for future readers.

The structure of the paper is well enough and the English are fine. After some modifications, the paper can be revised and published. 

Author Response

Point 1: The abstract should be modified. In the abstract and the body text, Some phrases are exaggerated. The authors claim that they develop a numerical model or method to evaluate the impact-resistant of a honeycomb structure, however they used the classical laminated theory. I couldn’t find novelty in model development however, the idea to use CNT combination with honeycomb is acceptable.

Response 1: Thanks for your recommendations. We have revised the abstract and the introduction section, and highlighted the application prospect of CNT combined with honeycomb in the protection field.

Point 2: The abbreviation word ‘FG-CNTRC’ in abstract should be defined first or use the full term.

Response 2: This comment was highly appreciated and we have added the definition of FG-CNTRC.

Point 3: They have used FEM software to evaluate the low-velocity response on the structure. This is a common mistake by many authors and to see the behavior of nanoscale materials FE method can not be reliable. This is because of continuum mechanics theory behind this method. It is not correct to use continuum mechanics that is defined for calculation of micro to meso scale. Many publications have repeated this mistake and due to a lack of experiments, this way is continuing. I strongly recommend you use molecular dynamic theories or nano-scale continuum modelling (NCM)  when you are studying CNT structure. However, they can use FE models to see only the trend of parameter variation and they need to mention this in their manuscript for future readers.

Response 3: Thanks for your recommendations. As you have mentioned, the method used in this research is more suitable for qualitative study on the variation trend of parameters. And we have added a relative description in subsection 4.1 of validation studies.

Reviewer 2 Report

The paper presents an analytical study of sandwich plates under impact. The authors use standard analytical equations which probably do not need to be completely re-printed herein, however this is not a problem and they are appropriately referenced. Currently there is no interpretation of the results. All results are simply stated - i.e. A has the smallest X, B has the largest Y. There is no interpretation of what these data mean for the design of such plates and for the research field more broadly. Meaningful interpretations should be added to the discussion and summarised in the conclusion - currently the conclusion simply re-states the results.

Author Response

Point 1: The paper presents an analytical study of sandwich plates under impact. The authors use standard analytical equations which probably do not need to be completely re-printed herein, however this is not a problem and they are appropriately referenced. Currently there is no interpretation of the results. All results are simply stated - i.e. A has the smallest X, B has the largest Y. There is no interpretation of what these data mean for the design of such plates and for the research field more broadly. Meaningful interpretations should be added to the discussion and summarised in the conclusion - currently the conclusion simply re-states the results.

Response 1: This comment was highly appreciated. We have added the interpretation of the results in the discussion part of each subsection 4.2 and the conclusion section.

Round 2

Reviewer 1 Report

The author's responses are satisfactory and now the manuscript can be published.

Reviewer 2 Report

The authors have addressed my comments